# Review of Advances in the Measurement of Skin Hydration Based on Sensing of Optical and Electrical Tissue Properties

**DOI:** 10.3390/s22197151

**Published:** 2022-09-21

**Authors:** Iman M. Gidado, Meha Qassem, Iasonas F. Triantis, Panicos A. Kyriacou

**Affiliations:** Research Centre for Biomedical Engineering, City, University of London, London EC1V 0HB, UK

**Keywords:** skin hydration, stratum corneum, skin optics, NIRS, electrical, biosensors, wearables

## Abstract

The presence of water in the skin is crucial for maintaining the properties and functions of the skin, in particular its outermost layer, known as the stratum corneum, which consists of a lipid barrier. External exposures can affect the skin’s hydration levels and in turn, alter its mechanical and physical properties. Monitoring these alterations in the skin’s water content can be applicable in clinical, cosmetic, athletic and personal settings. Many techniques measuring this parameter have been investigated, with electrical-based methods currently being widely used in commercial devices. Furthermore, the exploration of optical techniques to measure hydration is growing due to the outcomes observed through the penetration of light at differing levels. This paper comprehensively reviews such measurement techniques, focusing on recent experimental studies and state-of-the-art devices.

## 1. Introduction

The fundamental sub-layer of the skin when measuring skin hydration is known as the stratum corneum. The cells within this layer are embedded in a lipid-filled intercellular matrix, which allows for the skin surface to have waterproof properties. These lipids within the intercellular matrix play an important role in the maintenance of the barrier function and the control of transepidermal water loss (TEWL). Transepidermal water loss is a widely used method of measuring skin barrier function due to its objective nature. The preservation of water in the stratum corneum layer is attained through the barrier formed by the intercellular lamellar lipids and hydrophilic nitrogenous compounds that have the capability of holding moisture, a key example being natural moisturizing factors (NMFs). Overall, the principal factors that contribute to hydration level changes in the stratum corneum are water delivered from the epidermis, perspiration causing the evaporation of water from the skin surface and the water-holding capacity of the stratum corneum [1].

Skin hydration refers to the water content present within the cells. An increased intake of water or the use of a topical hydrator permeates the cells with water and improves the ability of the skin to absorb moisture. On the other hand, skin moisture is related to trapping the skin’s natural oils via the lipid cells to build its protective barrier and prevent water loss and dry skin. Moisturisers reduce the evaporation of water, thus minimizing TEWL. Skin moisture is directly related to conditions and diseases that cause dry skin, such as eczema. Lastly, body hydration as a whole refers to the overall body water content and encompasses the processes of dehydration and rehydration, commonly estimated through alterations in body mass and the tracking of fluid intake and losses [2,3].

The loss of water from the body in terms of management can be separated into sensible and insensible fluid loss. Sensible fluid loss relates to outputs of the body’s excretory pathway. These losses are apparent to the senses and therefore can be easily measured. Sensible loss examples include urine and faeces. On the contrary, insensible, or non-sensible, fluid loss is not easily recognised by individuals and is thus difficult to accurately measure. This type of loss has been found to increase in the presence of a disease that causes a rise in diffusion from the skin or lungs. Typical examples of insensible fluid loss are water released from the body during perspiration and respiration [4].

There are many different uses and motivations for the need to measure an individual’s hydration level [5,6]. An important motivation is hospital monitoring in terms of both paediatric dehydration assessment and fluid management, and dehydration monitoring in the elderly. Dehydration occurs when the output of bodily fluids is greater than the input. Infants are at a higher risk of dehydration due to their higher metabolic rates and higher water requirements per unit of weight than the average adult. The standard measurement method for dehydration diagnosis is the comparison of body weight before and after rehydration; however, the National Institute for Health and Care Excellence (NICE) have developed guidelines based on the assessment of clinical symptoms to detect and assess infant dehydration and shock [5]. Currently, there is no device with the capability to provide a direct measure of hydration in hospitalised infants and children.

On the contrary, elderly individuals are more prone to dehydration due to a higher susceptibility to disease states and physiological changes due to ageing. Moreover, the total water content in the body sees a reduction of 10–15% in the elderly, therefore making them more vulnerable to small fluctuations in water volume. The standard measure of dehydration in the elderly is fluid intake monitoring, since there is no basic, non-invasive method or device for measuring their hydration levels [7]. As individuals age, thirst becomes a poorer indicator for our fluid requirements. Therefore, thirst mechanisms are suggested to not always be a good indicator of dehydration. The intake of water relieves the sensation of thirst prior to the fluid replacement of the body being attained. This is also relevant in terms of athletes, as insufficient hydration can lead to impaired performance and poor body temperature and sweat regulation [8].

Another approach to the importance of skin hydration measurement techniques is the use in cosmetics [9,10]. Skin care companies have developed hydrators and moisturisers for both general usage and to aid individuals with skin conditions and diseases. Such skin conditions include eczema, psoriasis, xerosis and sun damage, with moisturisers and topical treatments aiming to reduce the skin effects of these conditions by providing a protective barrier to reduce water loss. Individuals suffering from eczema have a damaged skin barrier, therefore increasing their skin sensitivity to allergens and irritants and reducing the ability to retain water. Skin moisture can be measured with methodologies such as visual analysis, TEWL measurements or electrical-based techniques.

The measurement of skin hydration can also be beneficial for real-time monitoring in sports and fitness devices. The use of wearable devices that can monitor this measure would allow for the enhancement of an athlete’s hydration strategy and the optimisation of their performance through tracking and gaining an understanding of their personalised hydration needs. The typical sensors in these fitness-related devices are typically based on measuring the changes in the electrolytes in sweat over time or they can use electrical or optical sensors to measure hydration at a cellular level. Additionally, the need for portable devices is increasing significantly due to ease of use without specialist or hospital requirements [11]. Conducting research on the development of devices that allow for the measurement of skin hydration builds on the limited state of the art currently available within this area.

There are multiple techniques that can be used for the assessment of skin hydration [12]. Methods based on the electrical properties of tissue and water (henceforth termed as “electrical methods” or “electrical sensing” in this paper), such as the measurement of skin capacitance and conductance levels, are based on the concept that alterations in the electrical properties of the stratum corneum layer signify differing skin hydration levels. An alternative sensing approach involves interrogating the optical properties of tissues (henceforth termed as “optical methods” or “optical sensing” in this paper), including the use of near-infrared spectroscopy (NIRS). The latter involves the analysis of spectra within the near-infrared wavelength range, since intensities in the NIR spectrum of the skin consist of absorption bands directly related to its water content.

The aim of this paper is to comprehensively review the recent advances in developed sensors and wearable devices that utilise electrical and/or optical techniques to non-invasively measure skin hydration levels. The concept of combining techniques for multi-modal properties will be discussed and comparisons will be drawn on the advantages and disadvantages of both methodologies. In this review, there will primarily be a focus on experimental studies and a coverage of the current state of the art within the area of non-invasively measuring skin hydration. As mentioned, the motivation of this paper is to focus on the differences in measurement techniques in relation to accuracy, both in isolation and when combined. The idea of these devices being wearable will aid the real-time usage and provide a more portable and widely used device for use in various applications relating to skin hydration. Figure 1 illustrates a timeline detailing the chronological evolution of the fundamental techniques used in the measurement of skin hydration [13,14].

## 2. Methodology

Direct searches for identifying literature were made on PubMed, Google Scholar, Science Direct, City, University of London Library and MDPI. The stationary words used for the search criteria included “(skin hydration) AND (optical)” or “(skin hydration) AND (electrical)”. Additional key words such as “skin moisture”, “skin water content”, “NIRS”, “skin conductance” and “skin impedance” were also used within the search criteria. The inclusion criteria also comprised studies that involved in vitro and in vivo experimentation, with only human adults being used as the participant criteria for these studies. Furthermore, all publication types were considered with more focus on scientific experimental papers. Publications into skin hydration that did not include the use of an optical or electrical-based measurement technique were excluded. There was no specific time period restriction placed on the search; however, papers within the last decade were prioritised for a more profound analysis in this review paper. Results showing in a language other than English and also duplicate papers were omitted. These searches were performed continuously until 1 March 2022. Google Scholar yielded a high output of results, with 18,600 relevant publications related to optical measurements and 18,500 in relation to electrical measurements within the last decade. Thus, this was ordered in terms of date and only a subset of the more recent and relevant publications was selected. PubMed was the primary source used for the analysis of publications, with 121 optical related results and 358 electrical related results, again within the last decade, which were also screened for relevancy. The split of relevant publications is seen in the charts below in Figure 2 (bottom) and the leading publications and experimental studies used in this review paper are summarised in Appendix A. Figure 2 (top) illustrates a PRISMA flow diagram denoting the selection of studies and its progression from the databases.

## 3. Optical Sensing

### 3.1. Optical Imaging and Near-Infrared Spectroscopy

Optical imaging uses light to obtain and investigate images for medical applications. These images are typically produced within the ultraviolet to near-infrared wavelength region. Some common examples of optical imaging methods are optical microscopy, Doppler imaging and optical coherence tomography. Diffusive optical imaging utilises fluorescent-based techniques or near-infrared spectroscopy. This type of optical imaging refers to diffuse light that penetrates tissue at various projections [15].

Near-infrared spectroscopy (NIRS) covers the 780–2500 nm wavelength region of the electromagnetic spectrum and provides information on the perfusion of tissues. This is achieved through the measurement of light absorbance to monitor tissue oxygenation. This technique was initially used solely for the assessment of oxygen saturation in the brain but is more widely used for the analysis of other bodily tissues. An advantage of imaging within the NIR region is the high penetration depth into a sample in comparison to imaging in the mid infrared range. Although the sensitivity of this technique can be depicted as relatively high, there is the disadvantage of overlapping of broad absorption bands that would consequently require a mathematical data analysis. NIRS is based upon combination and overtone bands due to molecular vibrations, and as such, the bands visualised in the NIR region are characteristically broad and yield complex absorption spectra. To overcome this, multi-variate calibration methods are required to extract relevant information for subsequent analysis [16,17].

The typical NIR spectra of human skin consists of absorption bands that relate directly to water. The intensities within the absorption spectra have been found to be directly proportional to the skin water content. Thus, the given NIR spectra have the capability of differentiating various types of water in the stratum corneum layer of the skin. The absorption spectra of water in the near-infrared region consists of significant peaks, with two prominent peaks at 1920 nm and 1450 nm. The peak present at 1920 nm is the combination band from the OH stretch and HOH bands, whilst the peak at 1450 nm is the overtone band of the OH stretching. There are also additional weaker bands, for example around 1700 nm, due to the alkyl CH groups present in lipids and proteins within the skin. NIRS has the ability to directly detect water content in the skin through an analysis of the intensities of these combination and overtone bands within the NIR region [18].

### 3.2. The Use of Recent Optical Techniques in Skin Composition Studies

There are different optical techniques that have, more recently, been used to investigate the hydration levels of the skin. Some include confocal Raman spectroscopy, optical coherence tomography, speckle patterns analysis using optical tissue probing and optoacoustic monitoring. Another technique is near-infrared spectroscopy, which will later be discussed in more detail.

Ruini et al. [19] conducted an in vivo examination into the effect of moisturisers on human skin using both confocal Raman spectroscopy (CRS) and optical coherence tomography (OCT). The OCT device that was used was VivoSight (Michelson Diagnostics Limited, Maidstone, UK), which functions at a centre wavelength of 1305 nm and can achieve a penetration depth of 1.5–2 mm, producing two-dimensional cross-sectional images. Subsequent to analysis, the optical attenuation coefficient (AC) was found to be lower on more hydrated skin and would appear darker on OCT images in comparison to dry skin. CRS was performed on the gen2 Skin Composition Analyser (RiverD, Rotterdam, The Netherlands), which uses vibrational spectroscopy constructed on the principle of the inelastic scattering of light photons. This method gives the capability for analysis of molecular compositions and can therefore provide quantitative concentration profiles of the compounds in the stratum corneum, such as skin water content. The in vivo experiment involved 20 subjects with healthy skin, whereby measurements were taken before and after a two-week application of moisturiser on one forearm, and the other forearm acting as a control measurement. Following moisturisation, the results presented a positive correlation with water content along with increased epidermal thickness; however, a strong significance was not conveyed (*p* = 0.41). These results support the hypothesis that moisturisation primarily affects deeper epidermal layers more so than the SC layer. To conclude, it was established that the short-term application of moisturisers may be insufficient to achieve significant alterations in skin composition and morphology. This also suggests the need for multi-modal methodologies to accurately assess hydration levels of the SC.

The use of optical tissue probing with speckle patterns analysis can also be used as a technique to detect human skin hydration. This involves the temporal tracking of back-reflected speckle patterns while applying illumination and episodic vibrations. An optical system using this method was developed by Kelman et al. [20] and tested against the Corneometer^®^ (Courage and Khazaka, Cologne, Germany). Speckle patterns were obtained from an area of skin near to the centre of illumination to permit information from a smaller penetration depth, since it is associated with less-scattered light. The developed device is shown in Figure 3.

The in vivo experiment involved a range of dry to moist skin through the application of moisturiser on the volar arm, with measurements taken every 30 min across 3 h. The results showed that after the application of moisturiser, the optical signature displayed a significant decrease over time from the dry skin signature. Furthermore, it was also found that the higher the level of moisture, the faster the acoustic wave faded. With the calculation of the Pearson correlation coefficient, a negative correlation was identified between the trends of both instruments of R = −0.87. Unlike with the Corneometer, the developed optical system illustrated a large range of hydration readings and a high sensitivity for the identification of high hydration levels [20].

Perkov et al. [21] conducted studies on both gelatine tissue phantoms and human skin to investigate the monitoring of water content using optoacoustic (OA) methods. This particular technique is suggested to have a high resolution and contrast, due to the combination of ultrasound and optical methodologies, as well as a significant penetration depth for its probes, thus making it extremely appropriate for the monitoring of skin water content. The typical experimental setup for these measurements includes lasers emitted from a fibre bundle and an optical transducer connected to an oscilloscope, as seen in Figure 4. Through the analysis of the acquired OA signals, an evident second peak was discovered at a depth of 2 mm, which signified that the signals originate from subcutaneous tissue, since this tissue is found to have a higher water content than more superficial layers of the skin. The conducted in vivo experiments established the capability of the optoacoustic technique to effectively detect signals generated by water absorption in different skin layers [21].

In addition, Imhof et al. [22,23] conducted multiple in vivo experiments using optothermal methods to measure the hydration of the SC. A more novel technique they have used is the combination of opto-thermal radiometry and condenser-chamber TEWL. This allows for measurements of both SC surface hydration and evaporimetry, thus providing information on how the water diffusion coefficient depends on SC hydration.

### 3.3. The Use of NIRS in Skin Hydration Studies

In skin hydration studies, in vitro and in vivo hydration investigations are typically carried out on porcine skin and human participants, respectively. Qassem [13] conducted a study which investigated the properties of the stratum corneum in the NIR region of the EM spectrum and visualised the water characteristics in this layer using a spectrophotometer. The results of the in vitro experiments displayed two large peaks in both the water and porcine skin spectra at approximately 1450 nm and 1920 nm, with that of the porcine skin having an upward shift. This is due to porcine skin having a higher absorption coefficient for light than water. The existence of these peaks in the porcine skin spectra confirmed the presence of water in the samples. The results for the in vivo experiments showed peaks at 1450 nm and 1920 nm as expected, with the intensity at a lower magnitude than that of water. Beyond 1900 nm, the spectra exhibited high background interference due to the increase in penetration depth. This confirmed that deeper tissue is more saturated with water, thus having a higher absorptivity than more superficial skin layers. It was found that even though the probe being in direct contact with the skin allowed for water bands to be more clearly identified, this condition can lead to occlusion; therefore, non-contact readings are preferred. Moreover, a single fibre detector with a six-fibre source allowed for the most reliable and evident readings to be acquired.

A study with a similar methodology and measurement principle conducted by Kilpatrick-Liverman et al. [24] determined differences in the skin water content by measuring the absorption spectra using an NIR spectrophotometer with an optical fibre probe as well. The influence of relative humidity (RH) on skin water content was assessed via an in vitro experiment involving porcine skin equilibrated in desiccators containing saturated salt solutions at different concentrations. Gravimetric readings were then taken to measure the weight loss due to dehydration, which was utilised to calculate the percentage of water uptake. It was found that as the RH was decreased, lower water content was recorded in the skin. This result was established due to the area under the 1936 nm band being highest for skin that was equilibrated at 100% RH and lowest at 11% RH. Additionally, the effect of relative humidity (RH) and moisturisers on skin water content was evaluated by product application to participants in various in vivo experiments. In vivo experiments included studying the clinical effect of %RH changes via washing skin exclusively with water, the effect of humectants and the effect of a choline spray on skin water content. Readings were taken using the Skicon^®^ and NIR measurements and the obtained data analysed with paired t-tests comparing the changes from baseline readings. A direct correlation of R = 0.83 was observed between the %RH and NIR readings. It was concluded, from the results of the in vivo investigations, that products containing substances such as wax and oils smooth the surface of skin, which increased the beam penetration depth. Although these moisturizing products are implied to increase skin water content, they primarily increase the sampling volume due to this increased beam penetration.

Similarly, Arimoto and Egawa [14] conducted a study investigating measurements of non-contact skin moisture using NIR spectroscopy. In vitro experiments were carried out using porcine skin to explore the relationship between absorbance spectra peaks and water content. In vivo experiments collected diffuse reflected spectra using an optical fibre probe on skin. The measured spectra from these investigations were analysed with multiple linear regression (MLR) and partial-least squares (PLS) regression. These results were then compared to recordings obtained using a capacitance method. The results showed that the recorded measurement depth is highly dependent on the absorption of water, with deeper penetration present in spectral regions with weak water absorption. Moreover, this was found to be greater than the depth measured via the capacitance method. Correlations between water content and the second derivative were found to be highly significant at 0.99 and 0.98.

Furthermore, a subsequent study was conducted investigating the measurement of water content distribution in the skin [25]. This involved using confocal Raman spectroscopy to measure the vertical distribution of water in the skin. An estimation of the sensitivity of skin water content measurements were measured as well as an in vivo experiment to obtain NIR spectra. The skin water content was measured immediately after the application of a wet pad and again after 5 min. When the skin was moist, the water content was shown to decrease from the skin surface to a depth of 5–10 µm. Moreover, the water content at the skin surface was lower for the immediate recording. Beyond 5–10 µm, both water content recordings were almost identical. The imaging portion of the investigation involved an in vivo experiment that visualised the distribution of the skin water content at three different wavelengths. At a wavelength of 1300 nm, the pixel value was stagnant for 5 min after the removal of the wet pad. At 1462 nm, half the participants presented an increase up to 1 min, then a subsequent plateau. At 1950 nm, all participants displayed an increase up to 1 min, then again, a subsequent plateau. These results convey that a wavelength of 1950 nm had the highest sensitivity to changes in skin surface water content. Although there were absorption peaks at both 1950 nm and 1462 nm in the NIR spectral range, the pixel value variation at 1462 nm was smaller.

Alongside such studies, the effect of moisturisation on the skin is also focused on in the field of skin hydration measurements. A study by Qassem and Kyriacou [26] assessed the optical properties of the skin following water contact and the application of moisturiser using a spectrophotometer with a fibre optic probe. The resulting spectra as an average of all the participants recorded prior to water or moisturiser application displayed higher peaks of bands near 1450 nm and 1780 nm. This response was similarly seen when comparing individuals who frequently moisturise and do not moisturise. Results for participants with dry skin were observed to be most contradictory to those with normal skin, regardless of moisturisation. This suggests non-conformities in the barrier function characteristics of the skin and an increase in sensitivity with dry skin.

The advancement into the development of sensors contained within a built device is becoming more prevalent, as well as investigations into the use of multiple wavelengths when conducting optical related experiments. For example, Mamouei et al. [27] designed and developed a multi-wavelength optical sensor to measure dermal water content. The sensor consisted of two separate modules: the probe containing four LEDs, a photodiode and a transimpedance amplifier, and the main module. The main module encapsulated an analogue to digital converters (ADCs), current sources and also a connection to a USB port for data transfer via an Arduino microprocessor. Samples of porcine skin were prepared in an environmental chamber at 96% relative humidity and 25 °C for 48 h and then a desorption test was conducted that simultaneously measured the optical absorbance obtained from the developed sensor together with the sample weight. The designed prototype for the optical sensor used in this experiment is illustrated in Figure 5.

A further study was conducted using the same developed optical hydration sensor [28]. An in vitro experiment was performed on porcine skin to compare the developed sensor with a spectrophotometer to determine its performance and efficacy, with the reference being an electronic precision balance for gravimetric measurements. These weight readings served as a reference value for the water content in the sample, whilst the spectrophotometers reflectance spectra were used to benchmark the optical measurements. The porcine skin was segmented and inserted into an environmental chamber at 96% relative humidity and 25 °C. For conducting the measurements, both the probe from the developed sensor and the fibre optic probe from the spectrophotometer were placed and secured on to the skin surface. Both optical and weight recordings were obtained throughout a 3 h period. There was a high agreement between the absorbance results from the developed hydration sensor and the spectrophotometer, with both weight and absorption presenting a decrease as the water content diminished. Furthermore, optical measurements at the 1450 nm band displayed a higher sensitivity to water content variations. These comparisons indicate that the developed sensor exceeded the prediction accuracy of the spectrophotometer. Future investigations on the developed sensor would involve the addition of in vivo studies to assess its performance on human skin and examining the measurement accuracy when different skin types are considered.

Studies with developed optical sensors were found to be comparable to standard spectrophotometers, with prominent water peaks of their absorption spectra typically expressing in its correct wavelength regions. An increase in skin hydration led to a rise in the magnitude of these spectral peaks and direct correlations between water content and NIR readings. It has also been proposed that multi-wavelength near-infrared (NIR) optical sensors can exceed the accuracy of the standard NIR spectrophotometer with an increased sensitivity to minor alterations in the water content within the SC, in particular at higher wavelengths towards 1950 nm.

## 4. Electrical Sensing

### 4.1. Capacitance, Conductance and Bio-Impedance

Skin properties can be acquired by the analysis of its electrical parameters. A simplified lumped-impedance electrical model is considered, in which skin acts as a resistor placed in parallel to a capacitor. This model can be used to calculate the impedance values of the skin. Other electrical parameters, such as capacitance and conductance, can also be established via impedimetric devices, where electrodes are used with an applied alternating current [29].

The majority of the commercial devices used for assessing skin hydration are based on capacitive measurement techniques, with the gold standard device being the Corneometer^®^ (Courage and Khazaka, Cologne, Germany) [29,30,31]. The capacitance is recorded via two charged electrode plates creating an electric field, with the maximum charge produced being the measured capacitance value. A dielectric medium, such as the skin, being placed between the charged electrodes allows the capacitance to vary according to its permittivity. Water has a high dielectric constant of approximately 80, with a high dielectric constant relating to the ability of more charge being stored. Therefore, increasing skin water content is directly proportional to skin capacitance. Therefore, a highly hydrated stratum corneum would present a higher capacitance reading due to an increase in the induction of the dielectric constant. These electrical acquisitions relate to the overall skin hydration levels at the time of measurement; however, they do not consider all topographical surface measurements of the skin.

The skin conductance response, also referred to as the electrodermal activity, measures an increase in the activity detected by the sympathetic response system in response to a stimulus. This response acts as a method to measure the skin conductance, which varies according to moisture levels. Skin will momentarily alter its level of conductivity depending on the arousal of the internal or external stimuli. This measure increases linearly with SC water content [32].

Conversely, the electrical impedance of human skin depends primarily on temperature and on the transdermal voltage. The stratum corneum is isolated when conducting impedimetric measurements of skin hydration by the frequency of the alternating current being set to low values of around 0.1 to 1000 Hz, since impedance varies as a function of the frequency. With an increase in its water content, there is a relative increase in the dielectric constant and conductive pathways, which is more prevalent initially and then followed by a more subtle change over time. Thus, a decrease in impedance is observed as a function of SC water content [33,34,35,36,37].

### 4.2. The Use of Capacitance and Conductance in Skin Hydration Studies

A study that used capacitance as its measurement technique was conducted by Logger et al. [38], who investigated the anatomical site variation of water in the stratum corneum layer. The Epsilon (Biox Systems, London, UK), see Figure 6, was utilised for measurements, which records the skin capacitance and is essentially like the Corneometer device with multiple electrode pairs in comparison to the conventional single pair sensor. This design allows for multiple measurements to occur simultaneously. There were significant differences found in water content, with interindividual variations in terms of respective body locations, the largest being the cheek and smallest being the mid-calf region. The resulting values obtained from the Epsilon device were lower than conventional Corneometers outputs but followed a similar trend with *p* < 0.001.

In a more novel use of capacitive measurement principles, Flament et al. [39] developed a device named the Skin Hydration Sensor Patch (SHSP) that used a user’s smartphone to measure skin moisture in a wireless manner. It combined capacitive techniques and near-field communication (NFC) technology to allow for the self-testing of skin hydration via a probe attached to the back of a phone and recorded data were transmitted to a smartphone. A study was conducted to compare this device to the reference Corneometer, where the results presented a high correlation of r = 0.55 and *p* < 0.0001 between the methods. A further in vivo experiment consisted of participants with dermatologically assessed moderate dry skin, following the application of a hydrating Xanthane-based gel. The results showed that the values obtained from the SHSP had a strong positive correlation when compared to the Corneometer device output. In terms of the differences in hydration between differing skin sites, the face was found to have a lower hydration level than the forearm, with recordings from the face exhibiting a higher variability. Both skin sites presented similar trends subsequent to the application of the gel product, with hydration levels displaying a significant increase followed by a progressive decline in hydration across time. The hydration of the face expressed a higher amplitude, suggesting a differing requirement for hydration dependent on the skin region.

Another device that obtains capacitive measurements via an imaging technique called in vivo mapping is known as the SkinChip^®^ (L’Oréal, Paris, France). Developed by D Batisse et al., the device works by using the capacitance method to obtain components from the grey-level histogram of skin images to provide a non-optical representation of skin hydration [40]. Results from experiments displayed a linear correlation that is shown to be highly significant (*p* < 0.000) between the Corneometer recordings and the grey levels measured by the SkinChip device. Such devices have the ability to express the texture diversity of the skin surface, enabling the inhomogeneity of the skin hydration to be studied.

Skin conductance, in addition to transepidermal water loss (TEWL) and skin elasticity, can be examined as an index of the barrier function of skin. A study conducted by Nishimura et al. [41] investigated the effect of fine water particles on the moisture and viscoelasticity of facial skin. Skin conductance was recorded using the SKICON, TEWL by the Vapometer and skin distention by the Cutometer, at the cheek every 60 min. Participants were tested between three test conditions, each consisting of water particles of different sizes. It was found that the skin conductance of the stratum corneum was higher with smaller water particles. At a 120 min interval after spraying the water particles, the conductance was significantly increased in comparison with its baseline under all conditions. As these water particles were small in diameter and non-charged, they could permeate intercellular spaces in the epidermal layer through to the dermal layer of skin and maintain water retention.

Additionally, Andre et al. [42] developed a conductance-based device known as the Moisture Evaluator that is used to directly measure skin hydration during object manipulation. The probe of the device consists of gold-covered electrodes connected to a resistor–capacitor circuit to measure the hydration levels of skin based on the conductance principle. Comparisons showing correlations with the Corneometer will be referred to later.

Both skin capacitance and skin conductance have been presented to be sufficient in providing a good estimation on the detection of a high or low skin water content level. Multiple developed devices using the same methodological idea but different sensor design, such as the SKICON and Moisture Evaluator, have been found to follow similar trends and high correlations when compared to the Corneometer device. Capacitance-based devices, such as the Corneometer, were found to have a decreased sensitivity for monitoring high hydration values. On the other hand, conductance-based devices were found to have decreased sensitivity for low values. This can indicate that electrical techniques alone may not be sufficient in reliably measuring hydration levels.

### 4.3. The Use of Bio-Impedance in Skin Hydration Studies

The bio-impedance method can be used in skin hydration and barrier function experiments through its measurement principle of the electroconductivity of the skin. A study was conducted by Davies et al. [43], who produced a three-dimensional cell model, formed of layers of hexagonal cells acting as the stratum corneum above thicker layers of epidermal cells. The cells were suspended in a conducting medium and an electric field was then applied as well as an input varying sinusoidal voltage to obtain the impedance readings. Subsequently, the region of the model representing the stratum corneum was simulated at varying conductivities to measure the response to varying hydration levels. The results showed that at low frequencies, the conductivity had no effect on the overall impedance. However, when the frequency exceeded 100 kHz, the impedances were shown to diverge with the application of lower conductivity, causing a decrease in impedance. This model therefore supports the idea that increasing the hydration of skin can in turn reduce the skin impedance.

In addition, another study used the bioimpedance method to assess body hydration on cyclists [44]. A baseline measure of the participant’s bioimpedance was taken for result comparisons. Participants cycled for a few hours and had a low intake of water. Measurements were taken during hypohydration and after rehydration, in which the bioimpedance was remeasured. The results presented a correlation between hydration and the measured resistance. Sensitivity was shown up to around a 700 mL change in hydration across multiple tests.

Matsukawa et al. [45] conducted an in vivo experiment where skin impedance was measured on participants via nanomesh electrodes connected to an LCR meter to monitor skin hydration levels. The use of nanomesh electrodes has many advantages over standard devices that use rigid electrode probes, such as in the Corneometer, which require being in direct pressure with the skin. This can inadvertently alter hydration levels due to deformation of the skin surface. Additionally, the use of biocompatible electrodes that do not need pressure applied to the skin is considerate in terms of usage by individuals with skin conditions and diseases. The design for the electrode prototype in this experiment and technique via impedance measurements is shown in Figure 7. It was found that the recorded hydration levels showed a decrease as the skin impedance increased, with a negative correlation coefficient of R = −0.86, which supported the recognised hypothesis.

A study by Ali Imam Sunny et al. [46] had a different overall aim of developing a glucose monitoring system involving the use of a low-cost bioimpedance sensor developed to measure skin hydration. The experimental setup is illustrated in Figure 8. The sensitivity of the sensor was tested by measuring the impedance of differing concentrations of salt water solutions. The average impedance changes acquired over a frequency range of 30–50 kHz followed an exponential dependence with salt concentration. In addition, gelatine phantom measurements displayed an increase in impedance as the water content decreased over time. Only a less than 3% difference was identified in comparison to human skin, suggesting that the phantoms were suitable when testing correlations between impedance and water content, confirming that the sensor was effective in detecting minute skin hydration changes.

Somewhat different from a typical skin hydration sensor, Ameri et al. [47] designed a sensor similar to that of a tattoo, that had high stretchability while avoiding a loss of conductivity. It consisted of graphene electrodes that were tattooed directly to the skin surface and would remain functional for multiple days. The electrical impedance method was used to measure the humidity between two electrodes. It was found that the recorded results from this developed sensor were highly consistent with those obtained using silver–silver chloride gel electrodes.

Similarly, Shanshan Yao et al. [36] developed a skin hydration sensor that consisted of conformal silver nanowire electrodes. The sensor used the impedance method, with the capacitor comprising parallel electrodes inside a matrix. The sensor was packaged into a flexible casing that could be placed on the wrist and controlled by a microprocessor with a Bluetooth connection. This design enabled a wireless, low-cost, wearable device that could continuously measure skin hydration levels. The impedance was measured on artificial skin to establish the effect of external humidity on the skin impedance. The measurements confirmed that the sensor gave stable readings despite changes in the external surroundings. In further tests, the dielectric constant of the artificial skin when fully hydrated was measured. The impedance measurements exhibited a 0.62% increase as the water content decreased, with an exponential relationship when compared to the MoistureMeterD^®^ (MMD) device. Furthermore, in vivo experiments involved measurements before and after the application of lotion. As expected, a decrease in skin impedance was noted following the application of the lotion, thus indicating an increase in skin hydration.

Bioimpedance has been shown to be appropriate for the measurement of skin hydration due to its inverse relationship with the electroconductivity of the skin; thus, it has a negative correlation coefficient with skin impedance. However, techniques using bioimpedance have been shown to be unable to offer direct correlations with skin water content as the current within the skin can be influenced by changes in ion movements. Therefore, the idea of multi-modal techniques in combination with standard electrical methods should be explored as well.

### 4.4. Comparison of Electrical Sensing Devices

The Corneometer^®^ is a well-known device based on the principles of capacitance used to measure skin hydration levels. Its probe uses digital technology, allowing for increased stability and a lesser likelihood of interferences. On the other hand, the Skicon^®^ is a device based on the conductance method, where the conductance of a high frequency current is measured at 3.5 MHz. A visual comparison of the devices displays that the probe of the Corneometer^®^ has no galvanic contact of the electrodes to the skin, unlike the Skicon^®^ [48].

Clarys et al. [48] conducted experiments to draw comparisons between an analogue and digital version of the Corneometer^®^. Experiments were performed on cellulose filters and calibration filter pads immersed in various water-containing solutions. A significantly high correlation was found between the amount of water in the filter and the capacitance and conductance readings, with r being 0.89 and 0.99, respectively. A high correlation was also established between the dielectric constant and the electrical readings. An in vivo experiment was also carried out on different skin sites varying in hydration levels. Highly significant correlations were established between the devices, with an r of 0.98 to the analogue Corneometer^®^ and 0.97 to the digital version. Furthermore, an inverse relationship was shown between the hydration values and the capacitance and conductance recordings. This suggests a lower sensitivity of the Corneometer^®^ at high hydration levels.

Moreover, Logger et al. [38] investigated a conductance-measuring device known as the Epsilon™, similar to the Corneometer^®^ but consisting of multiple sensors. This device was utilised in a comparative study against the Corneometer^®^. As the Epsilon™ device has 76,800 sensors at a single probe, the advantage of multiple readings occurring simultaneously is exhibited. The hydration at the skin surface of five different body sites was recorded to explore the variation in the anatomical site on water content. The Corneometer^®^ displayed increased water content readings in comparison to the Epsilon™. However, both devices expressed similar results in that hydration levels increased at deeper layers of the skin. There was also a variation in surface water content at different anatomical regions, with the cheek and forearm displaying the highest water content levels and the calf showing the lowest. This validated that the skin water content measured in more hydrated skin layers will result in higher output values, as they comprise a thinner stratum corneum.

As previously stated, Andre et al. [42] developed a skin hydration device known as the Moisture Evaluator. This device was compared to a Corneometer^®^ to determine their correlations, which was found to be highly significant (0.2 N: R^2^ = 0.78, *p* < 0.01; 2 N: R^2^ = 0.83, *p* < 0.01) along with the determination coefficients; however, they displayed opposing sensitivity measures. There was found to be a decreased sensitivity with the developed Moisture Evaluator device for readings signifying dry skin. This is highly comparable to the suggestion that devices that utilise the conductance method have poor sensitivity for low levels of hydration. Additionally, the Moisture Evaluator demonstrated a higher sensitivity for higher levels of hydration than that of the Corneometer^®^, with responses still being exhibited at the point of saturation in the Corneometer^®^ output.

SkinUp^®^ is a device developed by Westermann et al. [49] which uses the impedance method to measure skin moisture and oil levels. A study was conducted to compare SkinUp^®^ to the Corneometer CM825^®^ to test the validity of the device. As moisturiser efficacy can be determined based on the skin’s electrical properties, this was also tested using the SkinUp^®^. The process in which this device utilises the impedance method followed the concept that hydrated, fat-free tissue has less electrical resistance or impedance, enabling the path for electric current. During the in vivo experiments, measurements using both devices were acquired on the forearm, cheek and forehead of the participants and these results were correlated via the calculation of Pearson’s correlation coefficient. Measurements were also taken after the application of two different formulations, being Lanette wax and distilled water with alcohol on a cotton pad. The results illustrated that highly significant correlations were found between both devices on all skin locations prior to any treatment application. The relative hydration level after the cotton pad application was 77% for the SkinUp^®^ device and 92.8% for the Corneometer^®^. After wax application, relative hydration was 54.4% with SkinUp^®^ and 25.3% with the Corneometer^®^. This expressed that the Corneometer^®^ was less sensitive to moisturizing formulations yet was seen to be higher with the application of water.

## 5. Commercial Skin Hydration Wearable Devices

Table 1 provides a summary of all available commercial skin hydration devices.

The desire of having sensors incorporated into wearables has become increasingly utilised for the monitoring of vitals, such as heart rate and sleep cycles. Different companies have recently used the understanding of how skin hydration can be accurately measured to import this into a wearable device. The most common measurement techniques are sweat monitoring via ions and electrolytes and measuring biomarkers from bodily fluids; however, optical and multi-modal techniques are becoming more explored in the wearable market.

Halo Wearables developed a non-invasive device, known as the Halo Edge, to monitor a user’s hydration levels at a cellular level by combining optical and impedance-based techniques. Using either of these modalities as a singularity is suggested by the developers to have the ability to provide a good estimate of hydration, but not a good measurement. The process of the device involves a learning and training method based on the individual’s independent hydrative state. The device, shown in Figure 9, is worn on the wrist and assesses dehydration measured from the individual’s sweat. The sensors in the device monitor the fluid levels in the blood and give a real-time measure of hydration based on a ‘Halo index scale’ ranging from 1 to 100. This index scale is sectioned into zones, each indicating the need for hydration. The target audience for the Halo Edge device is those training in sports; however, plans are being made to expand this to military and medical fields, as well as for personal monitoring [61,62].

Another company conducting this is Nix Biosensors, who worked with Harvard University to develop a single-use hydration sensor, primarily used for alerting low hydration levels in athletes. The feedback information is actionable and therefore allows the users to track their personalised hydration requirements. The developed device uses a sweat-based biometric sensor to monitor the changes in biomarkers present in body fluids [63].

Furthermore, hDrop is a developed wearable device that monitors the body’s electrolytes, temperature and hydration levels. The measurement process involves obtaining a voltage from the sweat response to then acquire the electrolyte concentration, since dehydration is directly related to a loss in electrolytes. The sensor conducts a small current through the skin surface in order to measure the electrical resistance of the detected electrolytes. The device can furthermore be linked to an app via Bluetooth and data can be stored for up to 4 h [64].

Sixty is another wearable device in a watch-like form that measures heart rate and hydration levels using optical spectrometry, and therefore consists of three LEDs and a photodiode. As well as providing a continuous and real-time measure of hydration levels, a companion app was developed to link one’s device to other smart devices. The app is able to provide advice on symptoms as well as being a store for the later analysis of measured behavioural patterns. The combination of hydration measures and heart rate monitoring via optical techniques allows for an increased accuracy in the provided readings [65].

Additionally, BSX Athletics developed a device known as LVL, which uses LEDs in the near-infrared wavelength range to measure heart rate and dehydration on the wrist. This optical method is used in combination with the measurement of sweat to provide real time feedback on bodily fluid requirements. The target audience for this device is athletes and it has been validated by over 250 volunteer athletes with comparisons made to the gold standards for hydration measurements of blood, urine and gravimetric techniques. Bluetooth is used to link the device to other smart devices, and it comprises an OLED screen for direct feedback to the user [66].

A team known as Sweatration investigated the issue where 80% of NCAA athletes had reported to be suffering from dehydration. They invented a hydration monitoring wearable that notified the user when dehydrated. The technique is based on a spike in sodium ions being directly related to high dehydration levels, therefore forming the methodology basis for the device. The spike in dehydration is monitored through the sodium ion conductivity levels. A working prototype has been developed but requires further refinements for market use [67].

## 6. Multi-Modal Measurement Techniques for Skin Hydration

The use of multi-modal techniques for the measurement of bodily parameters are becoming more valuable for their advantages. Such include an increase in the accuracy of readings, due to the combination of modalities providing a confirmation of the results. This acceleration in the application of multi-modal techniques highlights the increasing need for scientific advancements in this area to enable more reliable measurements. The ability to detect multiple parameters simultaneously with a single device has proved challenging for researchers, therefore driving the need to address this. Although there is not yet any combined analysis of multi-state variates within the experimental side of multi-modal sensing, efforts are being made by researchers to investigate this method of testing. A downside to multi-modal sensors opens the challenge of utilising a larger hardware circuit size and possible trade-offs in performance due to the micro-integration [68].

A study that used a multi-modal sensor in its methodology was conducted by Krishnan et al. [69]. They developed a device that introduced multi-modal sensors using both thermal transport and electrical properties to allow for measurements from sensitive areas of the skin. Analysis algorithms were utilised to provide the electrical impedance, temperature, thermal diffusivity and conductivity and heat capacity. In vitro studies were conducted on porcine skin soaked in saline water followed by independent measurements from the separate techniques. Temperature distributions of the porcine skin via infrared thermography presented that the level of hydration in the skin had a direct effect on its thermal behaviours. This is primarily sourced by secondary water in the epidermis, which also has a strong impact on the electrical properties. In vivo studies involve measurements taken from the volar forearm of healthy volunteers, with different locations of this area tested with or without the application of moisturisers with differing glycerine concentration levels. Each measurement would result in the output of impedance and thermal data, which would then be compared to the hydration readings obtained from a Corneometer device. Additionally, transepidermal water loss measurements were taken using a standard Tewameter device. Overall, the results conveyed a strong agreement between the developed multi-modal device and state-of-the-art techniques, suggesting a use for these device types clinically and for the real-time monitoring of hydration.

Another multi-modal approach was adopted by Cho et al. [70]. A chronic wound monitoring system was developed that could interface multiple signals, being voltage, resistance and capacitance measurements, using an integrated circuit. This gave the ability to acquire measurements of pH, temperature and humidity levels for diagnosis. In terms of the sensors used, pH was measured using a silicon substrate in combination with an electrode, an analogue voltage-type sensor was used to measure temperature and a capacitive-type sensor for measuring humidity. Conducted experiments included testing with buffer solutions ranging from pH = 4 to 10, using a heater with commercial temperature inputs, and a humidity chamber to acquire measured capacitance values. The results concluded that the developed multi-modal sensor system was able to deliver accurate and reliable diagnoses of chronic wounds at a lower cost and energy consumption, whilst still upholding a higher accuracy.

## 7. Discussion and Conclusions

The measurement of skin hydration levels is important in terms of different applications, whether it be in hospital monitoring for fluid management and dehydration assessment, or in cosmetic applications for personal usage or for individuals suffering from skin diseases. There is unfortunately no device that has been produced to give a direct measure of hydration in clinical environments. Therefore, the incorporation of this measurement into wearable devices can allow for a continuous and real-time monitoring system of skin water content, which can be used in hospital settings and in wellbeing and fitness. Different techniques can be used in the development of such sensors, in which both optical and electrical methods were explored within this paper.

Most commercial devices for measuring skin hydration are constructed utilising electrical-based measurement techniques, particularly the capacitance method. However, there are drawbacks that have presented through analysis of the use of these techniques. The Corneometer^®^ device has been found to have decreased sensitivity when measuring hydration levels of high value, suggesting that the capacitance method alone may not be sufficient in providing reliable results within the range of highly hydrated skin. Capacitance and conductance are considered to have an inverse relationship with one another, thus implying that conductance methods may be insufficient for measuring hydration at the other end of the spectrum, being low levels or dry skin [48,71].

Furthermore, there exists an issue of electrode-skin contact impedance for each electrode, which in a bipolar measurement is in series with the measured element (i.e., capacitance), generating a higher impedance measurement for smaller electrodes. Finally, measurements depend greatly on the contact quality, on the distance between the electrodes and on electrode material.

Bioelectrical-based techniques involve the injection of a constant current source and measurement of the resultant voltage. The resultant voltage can be measured by using the same pair of electrodes used for injection (bipolar) or a different pair of electrodes (tetrapolar). Bipolar electrode configuration suffers from the drawback of inaccuracies due to the addition of contact impedance to the resultant impedance of the tissue. As a result, this leads to measurement inaccuracies as one may not accurately determine the level of contribution to the total impedance due to interface impedances. Tetrapolar electrode configuration on one hand is insensitive to the changes in the interface impedance owing to the adaptation of different electrode pairs for current injection and voltage measurement. As a result, there is no current flow through the measurement electrode pair that could impair result accuracy. This configuration is particularly beneficial where very small and potentially micro electrodes are used in measurements. This owes to the fact that the current density developed by the injection pair at the electrode surface, as measured by the electrode radial distance, is inversely proportional to the electrode surface area. Hence, the smaller the electrode surface area, the larger the interface impedance. In addition, materials such as nanomesh electrodes have a larger area than plain metal and, therefore, a larger capacitance and in turn a smaller impedance. Although it is advantageous to use a tetrapolar connection for smaller sized electrodes, there is also the disadvantage of a more advanced circuit, which would increase the overall size of the device.

There has also been an increasing investigation into the use of optical techniques for the measurement of water bands within the absorption spectra of the skin. Research has found that the use of an optical sensor that consists of multiple wavelengths within the near-infrared region exceeds the prediction accuracy of a standard NIR spectrophotometer [13,28]. It expresses an increased sensitivity to variations in skin water content whilst being concurrent with the absorbance values obtained from the spectrophotometer. Selected wavelengths for these devices should correspond to the peaks in the water absorption spectra, especially near 1950 nm as this particular wavelength is suggested to have a high sensitivity to alterations in water content on the skin surface.

Optical techniques such as near-infrared spectroscopy (NIRS) provide a direct measure of the water content within the skin, thus allowing for an increase in the precision of the methodology, as well as a non-invasive and safer alternative to other techniques. Further limitations of NIRS in particular include the NIR region consisting of overlapping bands, meaning that bands can be easily influenced by other absorbers. Additionally, issues such as peak shifts due to temperature variations may exist. Another issue is selecting the appropriate bands (some bands such as 1950 nm are very difficult to find as LEDs, i.e., as discrete wavelengths, and having to use the entire spectrum would require a spectrometer, which is not very practical). Electrical methods on the other hand are unable to offer direct correlations between skin water content and skin conductivity, as the current exhibited can be influenced by alterations in ion movements. Furthermore, the electrical technique of using capacitance is achievable only with direct skin contact due to the occlusion of surface vapours. However, optical methods do not require direct contact to obtain readings. Electrical techniques have a higher usability for the analysis of deeper skin tissue, since methods using the measurement of conductance and capacitance can reach a depth of 40–100 μm from the surface of the skin, extending through to the upper epidermis layer. Both optical and electrical techniques have shown to be efficient in both their methods of measuring skin hydration, doing so in a non-invasive and achievable manner; however, a combination would be able to express the advantages of both techniques into a single modality.

In terms of multi-modal measurement techniques for skin hydration, the composite of different modalities allows for a further understanding of the skin’s distinct properties, whether it be optical, electrical or other methodologies. This can allow for the skin’s barrier function to be assessed in more detail from different perspectives and to achieve a multi-layer analysis due to the differing methods of measurements. For example, electrical techniques can provide measurements based solely on the water content and its effect on the conductivity of the skin’s surface, whereas the addition of optical techniques can use this information and further provide the water content through an intensity in the concentration of the detected water perspective. Furthermore, this combination could be superior to current state-of-the-art devices in skin hydration that currently use a single modality methodology in terms of the assurance of accurate measurements [72]. There are also disadvantages in using multi-modal techniques within a single measure device, with a prominent one being difficulties with post-acquisition analysis, such as when using multiple linear regression when analysing the results. In terms of skin hydration devices, the sensor would require calibration to allow for a baseline measure to be obtained prior to each reading. This measure would be user specific, which increases challenges in the accuracy of the measurements.

The use of wearable devices is becoming more prevalent among individuals, in terms of both healthcare monitoring devices and wearables in general. In particular, electrical-based methods are commonly used for the measurement of skin water content. However, these methods are highly dependent on the properties of the skin surface and its surroundings. More recent developments in wearable skin hydration devices also encompass other techniques, such as optical or sweat-based analysis, to achieve a more adaptable and user-independent device. A significant motivation for a more portable style wearable of this type is for use in sports and fitness applications. The requirements for the device involve needing to perform well against motion artefacts due to the high movement of athletes and to be lightweight. Due to the lightweight constraint, the device would need to encompass a small battery and therefore have a low power consumption; thus, as mentioned previously, the inclusion of tetrapolar electrodes are optimal. Examples of both state-of-the-art and novel commercial devices that have been found to successfully measure skin hydration have been outlined in Table 1.

To conclude, this paper has reviewed and analysed both experimental studies and state-of-the-art devices, encompassing both optical and electrical techniques into the measurement of skin hydration. Key motivations for the importance of measuring skin hydration were outlined as well as challenges that have been faced during the acquisition of these measures. With the current evolution of wearable devices, used in both healthcare and wellness applications, the innovation of a reliable skin hydration sensor/wearable is only a matter of time.

## Figures and Tables

**Figure 1 sensors-22-07151-f001:**
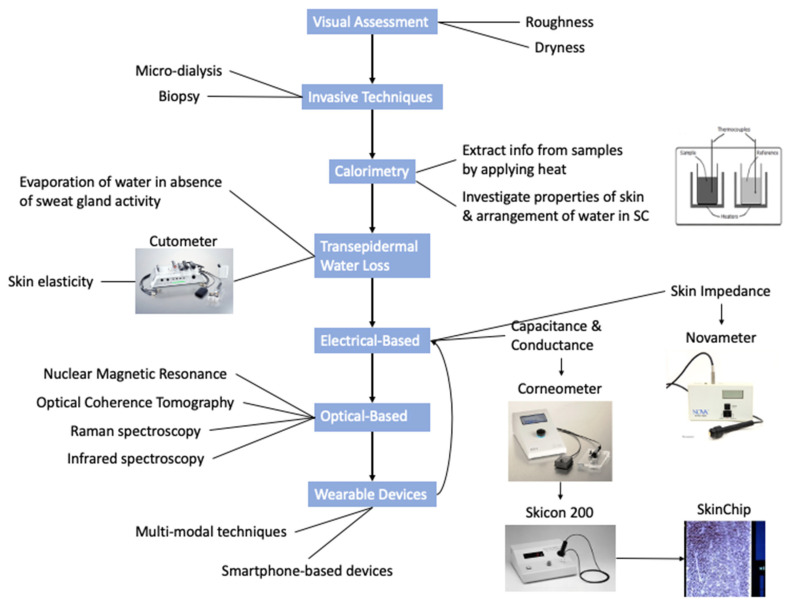
Timeline detailing evolution of skin hydration measurement techniques.

**Figure 2 sensors-22-07151-f002:**
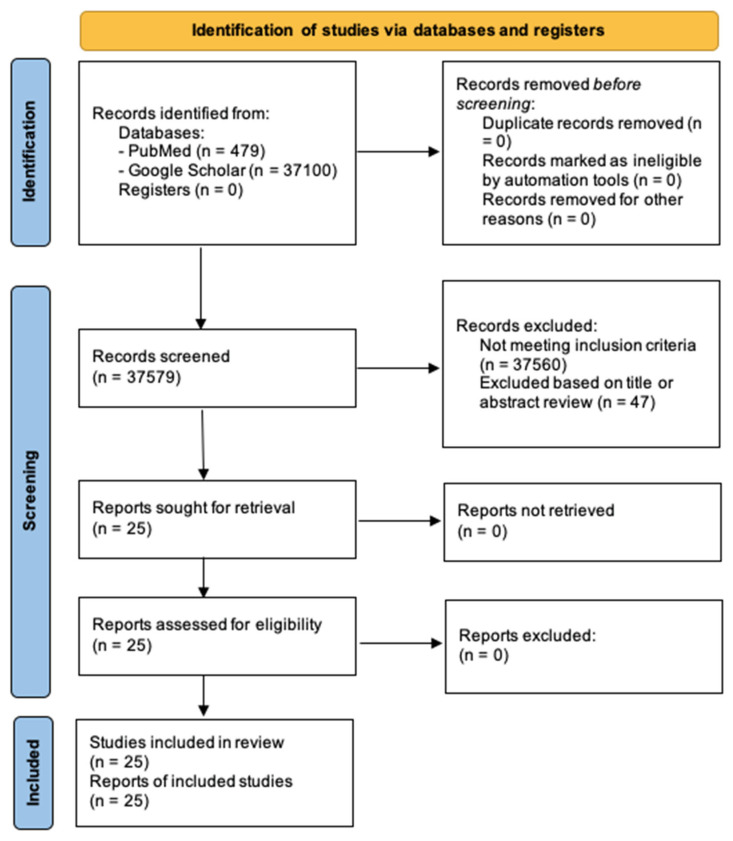
**Top**: PRISMA flow diagram of yielded database searches. **Bottom**: Bar charts to show split of relevant publications appearing from 2 primary search sources.

**Figure 3 sensors-22-07151-f003:**
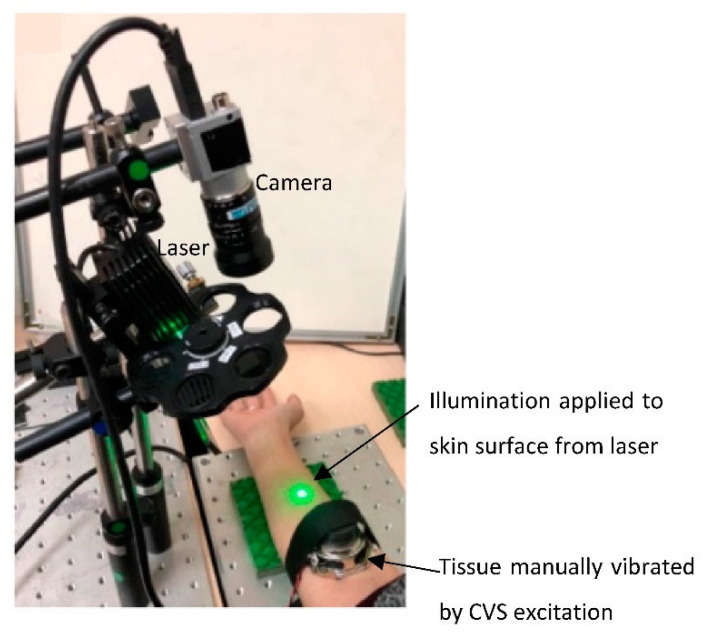
The experimental instrument for the optical setup, extracting the optical parameter to be compared to the Corneometer^®^ probe measuring capacitance. Modified from Kelman et al. [20].

**Figure 4 sensors-22-07151-f004:**
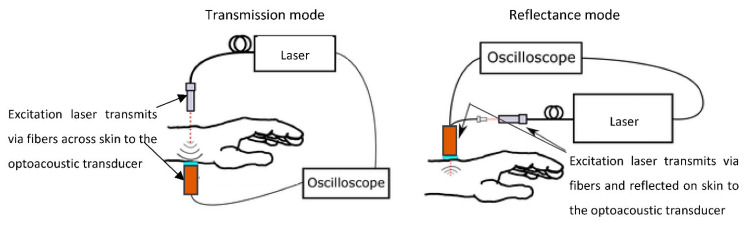
Experimental setup for OA measurements in the human wrist in the transmission mode and the reflection mode—modified from Perkov et al. [21].

**Figure 5 sensors-22-07151-f005:**
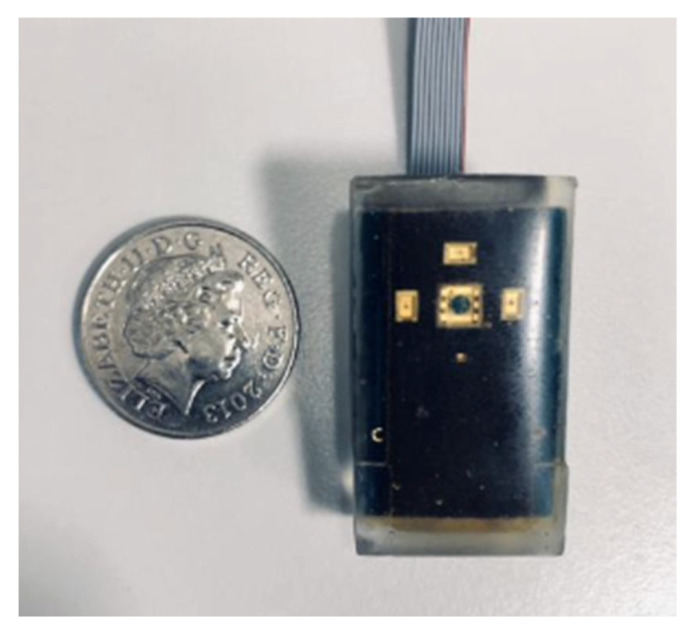
Developed prototype for optical sensor, consisting of four LED wavelengths and one photodiode [27].

**Figure 6 sensors-22-07151-f006:**
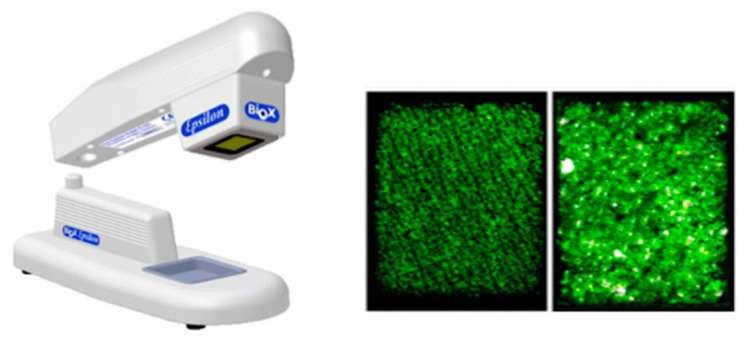
**Left**: The Epsilon instrument (Biox systems) with its in vivo parking stand; metal bezel can be seen on its head. **Right**: Typical contact image of the inner forearm skin and a contact image of the skin on the face with visible sweat gland activity—modified from Logger et al. [38].

**Figure 7 sensors-22-07151-f007:**
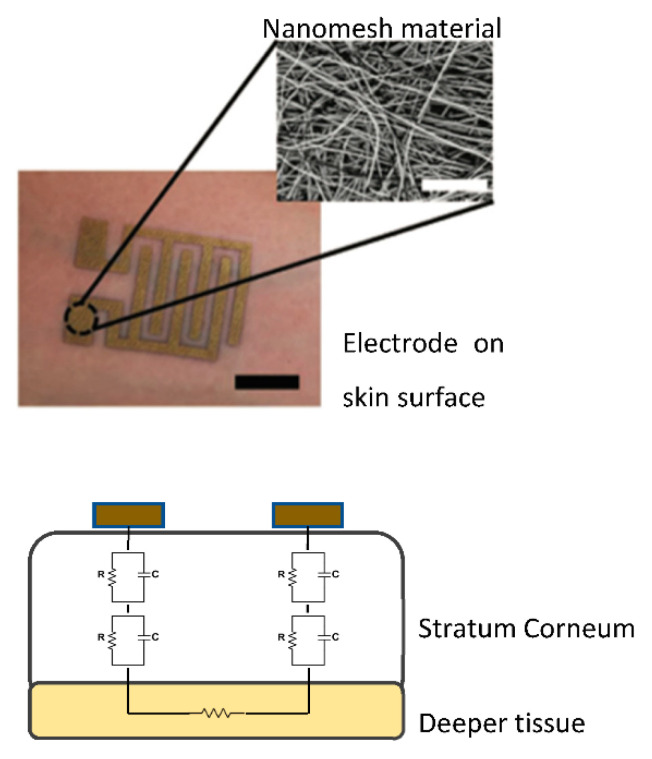
Measurement of skin impedance using nanomesh electrodes. **Top**: an optical photograph and SEM image of a nanomesh electrode pair attached to the skin. **Bottom**: the equivalent circuit of human skin. Modified from Matsukawa et al. [48].

**Figure 8 sensors-22-07151-f008:**
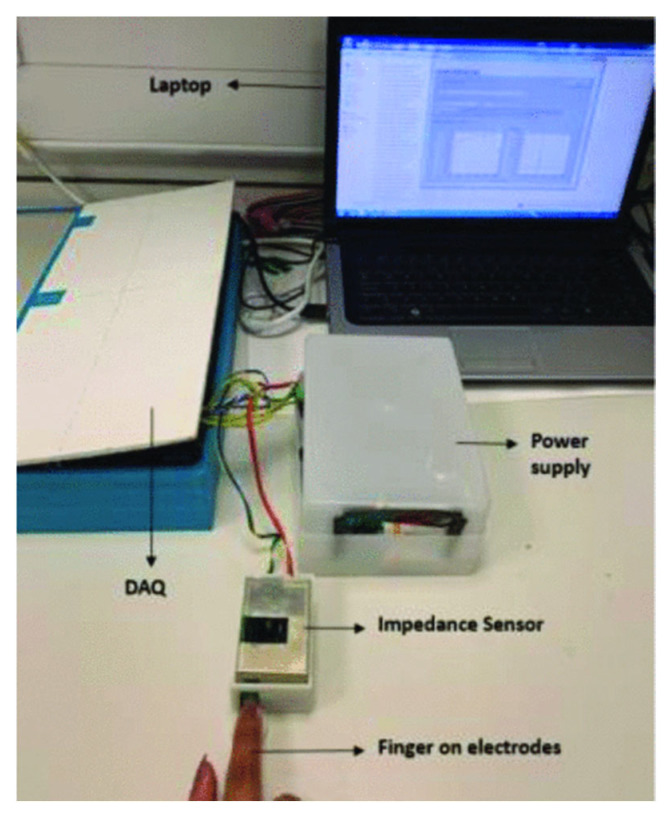
Laboratory setup for the sensing system [49].

**Figure 9 sensors-22-07151-f009:**
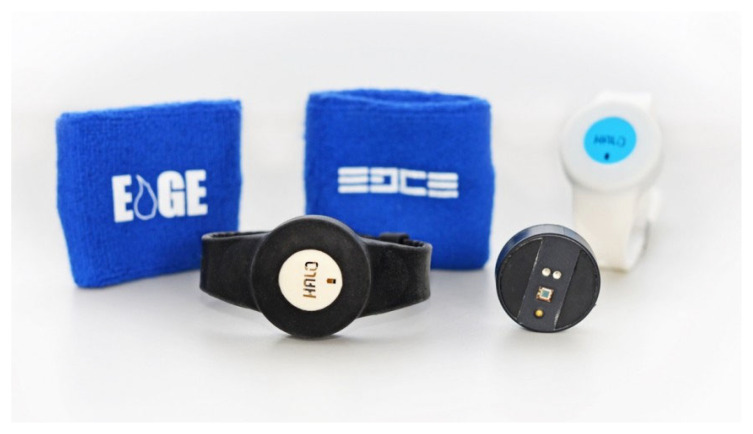
Halo Edge wearable hydration monitor, Halo Wearables [61].

**Table 1 sensors-22-07151-t001:** Commercial Hydration Devices.

Device	Company	Technique	Description of Measurement Process
Corneometer (CM825)	Courage and Khazaka	Conductance	Measures dielectric constant changes caused by skin surface hydration changing the capacitance of a precision capacitor. Correlation of 0.99 between analogue and digital versions [30]
MoistureMeterSC	Delfin Technologies	Capacitance	Applies an electromagnetic field to measure the skin’s dielectric constant. It is not affected by formulation electrolytes [31]
Skicon-200EX		Conductance	Measures the conductance of a single high-frequency current. Correlations of r = 0.98 with analogue Corneometer and r = 0.97 with digital Corneometer [32]
Nova Dermal Phase Meter (Novameter)	NOVA Technology Corp.	Impedance	Measures skin impedance to provide a non-invasive and objective measurement method to quantify the relative hydration of the skin [35].
SkinChip	L’Oreal	Capacitance	Based on an active capacitive pixel-sensing technology where the effective feedback capacitance is modulated by skin–sensor contact. Correlations of r = 0.69 versus Corneometer [40]
MoistureMap	Courage and Khazaka	Capacitance	Its sensor is based on capacitive-touch imaging technology to give graphical information on the skin hydration distribution and topography [50]
Surface Characterizing Impedance Monitor (SCIM)	U.S. Pat. No. 5353802	Impedance	Integrates readings taken at different frequencies of the applied alternating current to generate impedance-based values [51]
Tewameter	Courage and Khazaka	Open-Chamber Transepidermal Water Loss (TEWL)	Measures TEWL based on the diffusion principle in an open chamber to measure moisture at two sites. Correlation with gravimetric measurements, r = 0.7666 [52]
DermaLab	Cortex Technology ApS	Open-Chamber TEWL	Uses relative humidity and temperature sensors in an open chamber for a continuous readout of TEWL. ICC = 0.81 and r = 0.93 versus Tewameter [53]
Evaporimeter	ServoMed	Open-Chamber TEWL	Measures TEWL by estimating the vapour pressure gradient of water adjacent to the skin surface. Correlation of r = 0.97 versus Tewameter [54]
Vapometer	Delfin Technologies	Closed-Chamber TEWL	Monitors the increase in relative humidity inside the chamber to automatically calculate the evaporation rate. Correlation with gravimetric measurements, r = 0.763 [55]
Biox Aquaflux	Biox Systems	Condensed-Chamber TEWL	Measures TEWL using a condenser that continuously removes water vapour by conversion to ice [56]
Dermal Torque Meter (DTM)	Dia-Stron	Skin Elasticity	Induces a given amount of stress using a rotating disc on the skin, and then measuring the angular displacement of the skin deformation. Correlations of r = 0.54 with Cutometer for elastic deformation [57]
Twistometer	Dia-Stron	Skin Elasticity	Involves the induction of a given amount of stress using a rotating disc adhered to the skin, and then measuring the angular displacement of the resulting skin deformation [12]
Cutometer	Courage and Khazaka	Skin Elasticity	Uses a suction-based measurement principle using negative pressure and an optical measuring system. Correlations of r = 0.54 with DTM for elastic deformation [58]
Dermaflex	Cortex Technology	Skin Elasticity	Uses a suction-based measurement principle with a proportional strain method rather than the disproportional superficial system [59]
Raman Skin Analyser 3510	RiverD International B.V.	Confocal Raman Microscopy	A sample is illuminated for Raman scattering to occur, where the energy of the light is transferred to a molecule, exciting its vibrational modes to give a direct spectrum [60]

## Data Availability

Not applicable.

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
