# Peer review of "Review of Advances in the Measurement of Skin Hydration Based on Sensing of Optical and Electrical Tissue Properties"

_sensors, 2022, doi:10.3390/s22197151_

Round 1

Reviewer 1 Report

Gidado et al. in the manuscript reviewed measurement of skin hydration by using optical and electrical properties of tissue. The manuscript is well written and useful. I recommend for publication after revisions. I have the following comments and suggestions:

1.     Number the references in the order they appear in the text.

2.     Keep a space between a number and unit.

3.     Many sentences in Tables 1 and 2 are copied and pasted. They need to be paraphrased as it constitutes plagiarism. Also, refer the tables in the text.

4.     There should be a period after et al, e.g,“et al.”

5.     Please check the references very carefully. References 16 and 62 appears the same.

Author Response

The authors would like to thank the editor and reviewers for their valuable suggestions and comments. We addressed the comments and amended the manuscript accordingly. Changes in the manuscript have been highlighted in red. Please find below the detailed responses to the reviewers’ comments.

Reviewer 2 Report

Below are my comments

Section 1

1. The first paragraph has several lines that are not cited and then it starts with citation 3. Maybe there was an error in formatting, but this should be corrected

2. I think in the introduction when you talk about the importance of measuring hydration you should talk about sensical vs non-sensical fluid loss and why individuals don't recognize non-sensical losses of water. You should also talk about how thirst mechanisms aren't always a good indicator of dehydration when you're making the case for why it's important to measure hydration status.

3. You should cite lines 75-79

Section 2

1. How were the additional words used in the search criteria? I can't figure it out since you state that there were stationary words.

2. What were the other inclusion/exclusion criteria? Did you limit to human subjects only? What about age of subjects? 

3. A description of what the searches yielded should be displayed in a CONSORT flow diagram.

Section 3

1. In section 3.2 you should use citations for lines 165-167

2. Lines 182-191- why is it necessary to summarize the results of the study? If this paper is about reviewing relevant technologies to measure skin hydration the results of the study for skin thickness don't seem relevant to the actual technology, unless you can tie it back to hydration.

3. Did any of the technologies mentioned in 3.2 measure accuracy of measurement of skin hydration?

4. In 3.3 you should have some sort of summary paragraph that summarizes the various studies that you described. 

Section 4

1. You mention multiple studies that compared their technology to a Corneometer and state that there was high correlation. Can you provide the ICCs. Also did those studies create a Bland-Altman plot, if not it would be worth mentioning that while there was high correlation, they did not check for agreement.

2. You should also have a summary paragraph in 4.2 and 4.3

Section 5

1. I love the table in section 5. Is there any information available as to the accuracy of these monitors?

Section 6

1. Why not merge sections 5 and 6? 

Sections 7 and 8 are great

Another comment that I have that will make it significantly easier for the reader is to have tables for sections 3 and 4. It would help if you provided ICC values for some of the studies that reported ICCs. 

Author Response

(The authors gave the same response as above.)

Round 2

Reviewer 2 Report

I appreciate the authors responding to my recommendations. I would also like to apologize to the authors for making them complete a CONSORT flow when I meant to say PRISMA flow diagram. Can you please complete a PRISMA flow diagram. Below is a link 

https://www.prisma-statement.org/PRISMAStatement/FlowDiagram 

Author Response

We would like to thank the reviewer for their response. The CONSORT flow chart has now been replaced with a PRISMA flow chart in the manuscript.